# Myeloid Cell Mobilization and Recruitment by Human Mesothelioma in NSG-SGM3 Mice

**DOI:** 10.3390/cells13242135

**Published:** 2024-12-23

**Authors:** Vadim V. Shindyapin, Ekaterina O. Gubernatorova, Ekaterina A. Gorshkova, Nelya R. Chicherina, Fedor A. Sysonov, Anastasia S. Yakovleva, Daria A. Bogdanova, Oleg N. Demidov, Mariya V. Samsonova, Vladimir P. Baklaushev, Gaukhar M. Yusubalieva, Marina S. Drutskaya

**Affiliations:** 1Engelhardt Institute of Molecular Biology, Russian Academy of Sciences, 119991 Moscow, Russia; ekaterina.gubernatorova412@gmail.com (E.O.G.); gorshsama@gmail.com (E.A.G.); fsysonov@yandex.ru (F.A.S.); yakovleva.anastasia2002@gmail.com (A.S.Y.); baklaushev.vp@fnkc-fmba.ru (V.P.B.); kakonya@gmail.com (G.M.Y.); 2Department of Immunobiology and Biomedicine, Scientific Center of Genetics and Life Sciences, Sirius University of Science and Technology, Federal Territory Sirius, 354340 Krasnodar Krai, Russia; nelyayakhina@mail.ru (N.R.C.); bogdanova.da@talantiuspeh.ru (D.A.B.); demidov.on@mail.ru (O.N.D.); 3Center for Precision Genome Editing and Genetic Technologies for Biomedicine, Engelhardt Institute of Molecular Biology, Russian Academy of Sciences, 119991 Moscow, Russia; 4Faculty of Biology, Lomonosov Moscow State University, 119234 Moscow, Russia; 5Institute of Cytology, Russian Academy of Sciences, 194064 St. Petersburg, Russia; 6INSERM UMR1231, University of Burgundy, 21078 Dijon, France; 7Pulmonology Research Institute, Federal Medical-Biological Agency of Russian Federation, 115682 Moscow, Russia; samary@mail.ru; 8Federal Research and Clinical Center for Specialized Types of Medical Care and Medical Technologies, Federal Medical-Biological Agency of Russian Federation, 123098 Moscow, Russia; 9Federal Center of Brain Research and Neurotechnologies, Federal Medical-Biological Agency of Russian Federation, 117513 Moscow, Russia

**Keywords:** tumor microenvironment, pro-inflammatory cytokines, tumor-associated neutrophils, tumor-associated macrophages

## Abstract

Malignant pleural mesothelioma is a neoplasm that is often detected late due to nonspecific symptoms. This study utilized NSG-SGM3 mice to examine interactions between a human-derived mesothelioma reporter cell line (MZT-Luc2-mCherry) and the host’s myeloid compartment. Tumor growth was assessed using optical tomography, while cytokine/chemokine production was analyzed via multiplex assay. Histological and immunohistochemical analyses validated the epithelioid mesothelioma phenotype. In vitro mesothelioma cells secreted factors associated with myeloid cell chemoattraction and functions supporting the previously reported myeloid-biased secretory phenotype. In line with this, post-engraftment analysis revealed increased neutrophil-like Ly6G+ populations and decreased Ly6C+ inflammatory monocytes in the blood of tumor-bearing mice. Significant Ly6G+ cell infiltration was observed in the tumor, while CD11b+ myeloid cells were localized primarily in the tumor periphery. Tumor lysates showed increased levels of neutrophil chemoattractants and G-CSF, suggesting a previously not reported role of neutrophils in mesothelioma progression. This novel model provides a platform for studying mesothelioma–host interactions, focusing on the myeloid compartment. It may also serve as a tool to facilitate the development of new therapeutic strategies targeting myeloid cell-mediated mechanisms in mesothelioma.

## 1. Introduction

The World Health Organization estimates that approximately 30,000 people die from mesothelioma annually worldwide, with the highest rates observed in countries with historical asbestos use [1] and a global incidence that continues to rise despite efforts to limit asbestos exposure [2]. Malignant mesothelioma is an aggressive cancer primarily affecting the pleural and peritoneal cavities, with asbestos exposure being the primary risk factor, accounting for 80% of cases, although other factors such as radiation exposure and genetic predisposition have been implicated.

The pathogenesis of mesothelioma involves complex molecular mechanisms, including chronic inflammation, oxidative stress, and genetic alterations [3]. Recent genomic studies have identified frequent mutations in *BAP1*, *NF2*, *LATS2, SETD2*, and *TP53* genes, providing insights into potential therapeutic targets [4]. The long latency period between asbestos exposure and disease onset, typically 20–50 years, presents significant challenges for early diagnosis and intervention [5].

Mesothelioma is categorized into three main histological subtypes: epithelioid, sarcomatoid, and biphasic, as defined by the 2021 WHO Classification of Thoracic Tumours [6]. The epithelioid subtype, accounting for about 60–70% of cases, generally has a better prognosis compared to other subtypes. The sarcomatoid subtype, comprising 10–20% of cases, is associated with a more aggressive clinical course and poorer outcomes. The biphasic subtype, representing about 20–30% of cases, has an intermediate prognosis [7].

According to ESMO guidelines, mesothelioma diagnosis requires clinical, radiological, and pathological assessments [8]. Treatment includes surgery, radiation, and systemic therapies, with promising advances associated with immunotherapy. NCCN guidelines recommend immune checkpoint inhibitors as the first-line therapy for unresectable disease [9]. However, prognosis remains poor, with median survival of 12–16 months post-diagnosis.

Recent research has highlighted the critical role of the tumor microenvironment, particularly myeloid cells, in mesothelioma progression and patient outcomes. Myeloid cells, including tumor-associated macrophages (TAMs), myeloid-derived suppressor cells (MDSCs), and neutrophils, significantly influence mesothelioma development. Lievense et al. (2016) demonstrated that TAMs constitute a major component of the mesothelioma immune infiltrate [10]. These cells predominantly exhibit an M2-like phenotype, associated with tumor promotion and immunosuppression.

The presence of potential suppressor cells in mesothelioma correlated with poor prognosis. A study by Khanna et al. (2018) found that elevated levels of circulating potential suppressor cells in mesothelioma patients were associated with decreased overall survival [11]. Furthermore, they showed that potential suppressor cells may inhibit T cell responses, contributing to tumor immune evasion [12].

Neutrophils play a significant role in mesothelioma progression. Kao et al. (2013) reported that a high neutrophil-to-lymphocyte ratio in peripheral blood is predictive for poor outcomes in mesothelioma patients [13]. This finding suggests that systemic inflammation, partly mediated by myeloid cells, may influence disease progression. Neutrophils tend to accumulate in the tumor microenvironment of mesothelioma patients, contributing to disease progression and immunosuppression. The directed activation of neutrophils towards a suppressive phenotype leads to overall immunosuppression as these tumor-associated neutrophils inhibit T cell and NK cell functions, produce chemokines (CCL2, CCL3, and CCL5), and promote tumor growth and angiogenesis [14].

Therapeutic strategies targeting myeloid cells in mesothelioma are currently under investigation. Magkouta et al. (2021) reported promising results from a preclinical study using a CSF1R inhibitor (BLZ945 by Novartis Pharmaceuticals, Basel, Switzerland) to deplete TAMs in combination with immune checkpoint blockade [15]. This approach showed improved anti-tumor responses compared to checkpoint inhibition alone, highlighting the potential of myeloid cell-targeted therapies.

Preclinical models of malignant mesothelioma are critical for elucidating disease mechanisms and testing novel therapeutic strategies [16]. In vitro models, such as cell lines and organoids, already provided initial insights into mesothelioma biology, but they often fail to recapitulate the complex tumor microenvironment and immune interactions observed in patients [17]. The availability of genetically engineered mice which may provide a comprehensive platform for studying mesothelioma in vivo allowing observation of tumor growth, metastasis, and treatment responses remains limited [18]. To address these limitations, humanized mouse models emerged as a promising approach. Among these, the NSG-SGM3 mice support human immune cell engraftment [19,20,21,22], enabling studies of mesothelioma/immune system interactions and bridging the gap between animal models and clinical specimens [23,24]. These mice allow evaluation of patient-derived xenografts [25,26], drug screening for immunotherapeutic approaches [27,28], and research on various aspects including drug resistance, extracellular vesicles, microbiome, and tumor heterogeneity [29,30,31,32]. This platform may advance personalized therapeutic strategies in mesothelioma treatment [33].

The present study specifically addresses the residual activity of mouse myeloid cells without engrafting human bone marrow or immune cells. Such an approach focuses on baseline interactions between murine myeloid cells and human mesothelioma cells in the absence of a fully humanized immune system [34,35].

## 2. Results

### 2.1. Generation of Primary Mesothelioma Reporter Cell Line

The primary mesothelioma cell line was isolated and established (Figure 1A) from a pleural biopsy sample obtained from a patient diagnosed with the epithelioid histological subtype of malignant pleural mesothelioma. The primary cultures demonstrated high proliferative activity and maintained homogeneous morphology through multiple passages, with a 100% success rate in cryopreservation and revival.

To enable in vivo tracking and quantification, the primary mesothelioma cells were transduced with a bicistronic lentiviral vector encoding both mCherry fluorescent protein and Luc2 luciferase (Figure 1B). The resulting MZT-Luc2-mCherry cell line exhibited stable expression of both reporter genes. Flow cytometry analysis confirmed high expression of mCherry, with peak fluorescence emission in the 550–650 nm range (Figure 1C,D). In line with previously published data on cell morphology, we detected both spindle-shaped [36] and anchorage-independent growth (anoikis resistance) [37].

To provide context for the unique cytokine profile of mesothelioma, glioblastoma and acute myeloid leukemia cell lines were compared, representing distinct cancer types with specific cytokine signatures. This comparison allows for the identification of mesothelioma-specific cytokine context, highlighting its distinct immunological features. Cytokine profiling of the mesothelioma cell line suspension (Appendix A) revealed significantly elevated production of G-CSF, GM-CSF, IL-8, CXCL1, and IL-1α compared to glioblastoma and acute myeloid leukemia cell lines (Figure 1E). These cytokines may contribute to the unique tumor microenvironment and myeloid-biased recruitment as previously reported for mesothelioma [34]. The elevated levels of G-CSF and GM-CSF are particularly interesting, as they may suggest a potential mechanism for the recruitment and expansion of myeloid cells, in particular neutrophils, within the tumor microenvironment. Moreover, the high production of IL-8 and CXCL1 indicates a strong potential for neutrophil chemotaxis which aligns with our subsequent observations in the xenograft model. Interestingly, chemokines involved in monocyte/macrophage recruitment and mobilization such as CCL2, CCL3, CCL4, and CCL7 were downregulated in the mesothelioma cell line compared to two other tested cell lines.

### 2.2. Histopathological and Immunohistochemical Analysis of the Engrafted Mesothelioma

To evaluate the in vivo growth features and confirm the histopathological features of mesothelioma, we subcutaneously injected the MZT-Luc2-mCherry cell line into the flank region of NSG-SGM3 mice. Data from histopathological and immunohistochemical analyses of the engrafted tumor were characteristic for epithelioid mesothelioma (Figure 2A) and were consistent with the histological subtype of the primary tumor. The engrafted tumor retained key mesothelioma features, including strong positivity for Pancytokeratin (Figure 2E), Calretinin (Figure 2D), and WT1 (Appendix A), as well as typical expression patterns for mesothelioma Mesothelin (Figure 2F) and EMA (Figure 2C). Immunohistochemical analysis demonstrated strong and diffuse positivity for Pancytokeratin and Calretinin and revealed positive staining for Mesothelin, with a predominantly membranous–cytoplasmic expression pattern in the neoplastic cells. Moderate membranous expression of EMA was observed in the majority of tumor cells. Focal immunoreactivity for EMA and Mesothelin was also observed. The high proliferative index, as indicated by Ki-67 immunostaining (Figure 2B), with positive nuclear staining in over 60% of tumor cells, further confirmed the aggressive nature of the established tumor model. These findings indicated that the engrafted tumor preserved both the histological appearance and fundamental properties of the original mesothelioma.

### 2.3. Characterization of the Residual Myeloid Compartment of NSG-SGM3 Mice

To establish a suitable in vivo model for studying mesothelioma growth and its interactions with the host myeloid cells, we evaluated the frequency of these cells in the blood of naïve NSG-SGM3 mice as compared to C57BL/6 mice. The NSG-SGM3 mice which expressed human IL-3, GM-CSF, and SCF as transgenes [35] were selected as the recipients for the xenograft model due to their enhanced support of human cell engraftment and residual myeloid cell activity (Figure 3A). These mice lack T, B, and NK cells, making them useful for studying human tumor growth without activating adaptive immune responses or NK cell-mediated cytotoxicity. Flow cytometry analysis of peripheral blood from naïve NSG-SGM3 mice revealed significantly lower CD45+ cell counts in NSG-SGM3 mice compared to C57BL/6 mice (*p* < 0.0001) (Figure 3B). While no significant difference was observed in absolute CD11b+ cell counts (Figure 3C), NSG-SGM3 mice exhibited a higher percentage of CD11b+ cells within the CD45+ population (*p* < 0.0001) (Figure 3D). The observed shift in peripheral blood immune cell composition with a bias towards the myeloid compartment in NSG-SGM3 mice could create an environment that is more suitable for the engraftment and growth of human mesothelioma cells due to its myeloid-prone secretory phenotype.

Representative flow cytometry plots demonstrated distinct myeloid cell populations in C57BL/6 and NSG-SGM3 mice (Figure 3E,F and Appendix A). The unique immunological profile of NSG-SGM3 mice, particularly the altered myeloid compartment, provided a suitable environment for studying human mesothelioma xenografts and their interactions with the host myeloid system. The detection of human cytokines produced by tumor cells (Figure 1E) in this model may contribute to understanding of a complex cross-talk between tumor and host immune cells, revealing new aspects of mesothelioma progression and growth.

### 2.4. Establishment of Mesothelioma Xenograft Model in NSG-SGM3 Mice

To assess the engraftment efficiency and growth kinetics of the MZT-Luc2-mCherry cell line in vivo, NSG-SGM3 mice were injected s.c. with 2.5 × 10^5^ cells and monitored for tumor development using multiple imaging modalities over a 4-week period (Figure 4A). First of all, 100% engraftment of the MZT-Luc2-mCherry cells in NSG-SGM3 mice was achieved. Optical tomography measurements showed a steady increase in mean fluorescence intensity over 28 days (Figure 4B), which correlated well with tumor volume assessed by manual caliper measurements (Figure 4C,D). The absence of visible necrotic regions in the growing tumors indicated adequate vascularization and successful establishment of the xenograft model. Such a high engraftment rate and the consistent growth pattern suggest that the MZT-Luc2-mCherry cells are well adapted to the NSG-SGM3 microenvironment, providing a reliable model for studying mesothelioma progression and potential therapeutic interventions targeting the myeloid compartment. Additionally, bioluminescent imaging was performed following luciferin administration to assess metastatic spread. No metastatic lesions, however, were detected within the 30-day observation period.

### 2.5. Distribution of Myeloid Cells Following Tumor Engraftment

To investigate the systemic effects of mesothelioma on the host myeloid compartment, we performed flow cytometry analysis of peripheral blood samples from naïve and tumor-bearing mice. Flow cytometry analysis of peripheral blood samples of mice revealed significant changes in myeloid cell populations following mesothelioma xenograft establishment. Compared to naïve NSG-SGM3 mice, tumor-bearing mice showed a marked increase in the neutrophil-like Ly6G+ population and a decrease in the monocyte-like Ly6C+ population within the CD11b+ compartment (Appendix A). This shift in myeloid cell composition suggested a systemic impact of the growing tumor on the host immune system, potentially mediated by tumor-derived factors such as G-CSF and GM-CSF. It is important to note that while there is no cross-reactivity between human and mouse GM-CSF, such cross-reactivity does exist for G-CSF with implications for the interpretation of the results in human-to-mouse xenograft models [38].

Paired analysis of individual mice before and after tumor implantation demonstrated a consistent shift in these populations (Figure 5A). The significant increase in Ly6G+ cells (*p* < 0.0001) in the blood is likely due to active proliferation and mobilization of these cells rather than a relative increase caused by the depletion of Ly6C+ cells. Histological examination (Figure 5B,C) suggested an influx of polymorphonuclear cells within the tumor microenvironment and at its periphery, highlighting their critical role in the immune response to the tumor. This was further supported by the observed infiltration of Ly6G+ cells into the tumor tissue (Figure 5D). The expansion of the neutrophil-like population in both the blood and tumor microenvironment suggested a strong neutrophil-driven response to the growing mesothelioma, which aligns with previous reports of neutrophil involvement in mesothelioma progression [39]. This increase in neutrophil-like cells may contribute to an immunosuppressive microenvironment [40,41], potentially facilitating tumor growth in the absence of adaptive immune responses. The active proliferation and recruitment of Ly6G+ cells, rather than just a relative increase due to Ly6C+ cell migration, emphasizes the involvement of neutrophils in our model of mesothelioma progression.

Immunofluorescence staining of tumor sections revealed substantial infiltration of CD11b+ and Ly6G+ cells (Figure 5D). Interestingly, we observed a distinct spatial organization of myeloid cells within and around the tumor. The tumor core was predominantly infiltrated by Ly6G+ neutrophil-like cells, while the tumor periphery was surrounded by a dense layer of CD11b+ cells, which did not stain with Ly6G, presumably, monocytes/macrophages.

The distinct compartmentalization of neutrophils and monocytes/macrophages suggests that the tumor may actively shape its microenvironment by recruiting and positioning specific myeloid cell subsets to support its growth and survival, even in the absence of adaptive immune responses.

### 2.6. Cytokine Profiling of NSG-SGM3 Tumor-Bearing Mice

To characterize the in vivo cytokine profile associated with mesothelioma progression and its impact on the host myeloid compartment, we analyzed both human and murine cytokines in serum samples and tumor lysates from tumor-bearing NSG-SGM3 mice. Analysis of tumor-derived human cytokines in serum samples from tumor-bearing NSG-SGM3 mice revealed a significant level of IL-8 production (Figure 6A). IL-8 is a known pro-angiogenic factor and neutrophil chemoattractant that has been implicated in mesothelioma progression and poor prognosis [42,43,44,45]. Moreover, it plays a crucial role in mobilizing immature myeloid cells and enhancing the inflammatory response [46]. The presence of human IL-8 in the murine serum demonstrates the successful engraftment and metabolic activity of the human mesothelioma cells in this xenograft model consistent with in vitro data (Figure 1E). Murine cytokine profiling of serum samples showed upregulated production of cytokines involved in neutrophil differentiation and recruitment (G-CSF and GM-CSF); monocyte/macrophage mobilization and recruitment (CCL2 and CCL5); inflammation (TNF and IL-1β); and immunomodulatory cytokines (IL-10 and IL-15) (Figure 6D). The upregulation of murine G-CSF and GM-CSF in both serum and tumor lysates (Figure 6B, Appendix A) corroborates the observed expansion of the neutrophil-like population and suggests a potential feedback loop between the tumor and the host myeloid system. The simultaneous increase in pro-inflammatory (IL-1β, TNF) and immunomodulatory (IL-10, IL-15) murine cytokines indicates a complex immune response to the growing tumor, potentially reflecting attempts at both anti-tumor immunity and immune suppression within the limited innate immune repertoire of NSG-SGM3 mice. Consistent with cytokine profiling, real-time PCR data confirmed upregulation in the expression of *IL8*, *Cxcl10*, *Il1β*, and *Ccl2* in tumor lysates (Figure 6C).

## 3. Discussion

The present study demonstrates the successful establishment in NSG-SGM3 mice of a novel mesothelioma model using the MZT-Luc2-mCherry cell line derived from human pleural biopsy tissue. This model provides valuable insights into the interactions between mesothelioma cells and the host’s myeloid compartment, offering a useful platform for investigating tumor progression and searching for potential immunotherapeutic strategies. The NSG-SGM3 mouse model expresses human cytokines IL-3, GM-CSF, and SCF, which can interact with human tumor cells and murine cells, potentially affecting immune microenvironment. Cross-species differences may result in altered cytokine signaling, myeloid cell recruitment, and function [47,48]. In the future, introduction of humanized mouse models with reconstituted human immune system, including the adaptive immunity, may provide a more accurate replication of the disease microenvironment [49]; however, our model is useful for studying myeloid cell interactions and testing therapies specifically targeting the myeloid compartment.

Our findings reveal that mesothelioma cells exhibit a distinct cytokine and chemokine secretion profile, particularly including those associated with myeloid cell attraction and function both in vitro and in vivo (Figure 1E and Figure 6A). This observation aligns with previous studies highlighting the importance of inflammatory mediators in mesothelioma progression [50]. The increased production of neutrophil chemoattractants and G-CSF by tumor cells suggests a previously underestimated role of neutrophils in mesothelioma growth and progression, which warrants further investigation [14,51,52,53].

The observed changes in myeloid cell populations, specifically the increase in neutrophil-like Ly6G+ cells and the decrease in Ly6C+ inflammatory monocytes in the blood of tumor-bearing mice (Figure 5A), provide new insights into the systemic effects of mesothelioma on the host’s immune system. These findings are consistent with recent studies demonstrating the importance of neutrophils in tumor progression and metastasis in various cancer types [54]. This further resonates with the comprehensive reviews, which delineated the multifaceted functions of neutrophils within the tumor microenvironment [55,56]. Subcutaneous tumor implantation can trigger local inflammatory responses, leading to neutrophil recruitment. However, the mesothelioma cell line described here secreted high levels of neutrophil-specific cytokines, such as IL-8 and G-CSF in vitro (Figure 1E). In line with this, we demonstrated the elevation of human IL-8, as well as murine G-CSF and GM-CSF, in tumor-bearing mice (Figure 6A,B). The observed increase in tumor-derived factors may play a significant role in neutrophil mobilization. Neutrophils are known to contribute to tumor progression by promoting angiogenesis and suppressing anti-tumor immunity [41]. In mesothelioma patients, elevated neutrophil-to-lymphocyte ratios were associated with a poorer prognosis [57]. These findings indicate that neutrophil infiltration is primarily driven by tumor-intrinsic factors, although cells at the implantation site may also contribute.

In the present study, we reported infiltration of Ly6G+ cells within the tumor and the localization of CD11b+ myeloid cells in the tumor periphery. This observation is in line with a recent report summarizing the pro-tumorigenic functions of tumor-associated neutrophils in other cancer models [58]. The compartmentalization of myeloid cell subsets (Figure 5D) suggests the presence of functionally distinct niches within the mesothelioma microenvironment. The abundance of CD11b+ cells at the tumor periphery suggests a potential immunosuppressive barrier formed by myeloid cells, which may contribute to tumor immune evasion [59]. Recent studies have demonstrated that macrophages can adapt an M2 phenotype in response to tumor microenvironment signals, leading to an immunosuppressive milieu that facilitates tumor progression [60]. This “myeloid/monocyte shield” could impede the infiltration of any remaining innate immune cells and promote a pro-tumorigenic microenvironment. Neutrophils within the tumor may further positively affect tumor growth through the release of pro-angiogenic factors and matrix-remodeling enzymes [41].

The observed myeloid cell recruitment and secretory phenotype in our model have significant implications for immunotherapy development in mesothelioma, with recent findings in other cancer types demonstrating that TANs can exhibit both pro- and anti-tumor effects depending on the tumor microenvironment [61,62]. These innate capacities of neutrophils are being actively explored in innovative therapeutic approaches, such as engineering neutrophils to actively deliver drugs to malignant gliomas, utilizing their natural abilities for chemotaxis and phagocytosis to overcome barriers and enable targeted therapy [63]. Similarly, in gastric cancer treatment, neutrophils have been successfully employed as delivery vectors for Abraxane in combination with radiotherapy, where radiation-induced inflammatory factors guide neutrophils to tumoral sites and trigger neutrophil extracellular trap formation for enhanced drug release [64]. The recent study on combination immunotherapy in mesothelioma provides a relevant clinical context for our findings [65]. The authors reported on the long-term outcomes of combining ipilimumab (anti-CTLA-4) with nivolumab (anti-PD-1) in patients with unresectable malignant pleural mesothelioma. Our model of human mesothelioma engraftment in NSG-SGM3 mice demonstrates that myeloid-specific secretory phenotype observed in vitro translates in vivo to mobilization of myeloid cells to the tumor site and distinct spatial organization of Ly6G+ neutrophil-like cells and CD11b+ monocytes/macrophages within and around the tumor. In the future, it will be crucial to validate these findings in other preclinical models.

## 4. Materials and Methods

### 4.1. Cell Line Generation and Characterization

The primary mesothelioma cell line was established from a pleural biopsy of a patient diagnosed with malignant pleural mesothelioma. The tumor mass was mechanically dissociated to generate a single-cell suspension. Cells were cultured in treated 75 cm^2^ flasks in DMEM/F12 medium supplemented with 10% FBS and antibiotic–antimycotic solution. For initial primary cell culture establishment, 1 × 10^6^ cells were seeded, and subsequent passages were performed after achieving full monolayer formation. During the first week, cells were passaged weekly, followed by passages every two days thereafter. The cells were passaged 20 times before genetic modification, with approximately 2 × 10^6^ cells being harvested and 2 × 10^5^ cells reseeded at each passage. The cell line was then transduced with a lentiviral vector (pCDH-EF1a-Luc2-IRES-mCherry) encoding two reporter proteins: the fluorescent mCherry and the bioluminescent Luc2. Stably transduced cells were selected using a Sony SH800S (Sony Biotechnology, Weybridge, UK) cell sorter based on mCherry fluorescence. The sorted population was maintained for five additional passages under the same culture conditions. The success of the transduction was monitored by fluorescence microscopy (CELENA^®^ S Digital Imaging System l, Logos Biosystems, Anyang-si, Republic of Korea) and by flow cytometry to confirm reporter gene expression.

### 4.2. Mice

The NSG-SGM3 mice (NOD.Cg-*Prkdc^scid^ Il2rg^tm1Wjl^* Tg(CMV-IL3,CSF2,KITLG) 1Eav/MloySzJ, JAX stock #013062) were obtained from The Jackson Laboratory (Bar Harbor, ME, USA). This strain was originally described in Wunderlich et al., 2010 [66] and Billerbeck et al., 2011 [67]. Both NSG-SGM3 and C57Bl/6 mice were maintained at the Almazov National Medical Research Center. In vivo imaging was carried out at the Animal Facility of the Center for Precision Genome Editing and Genetic Technologies for Biomedicine, EIMB RAS. All animal procedures were performed in accordance with Russian regulations on animal protection and approved by the local Ethics Review Committee at EIMB RAS (Protocol No. 1 from 4 March 4 2024).

### 4.3. Tumor Implantation and Monitoring

MZT-Luc2-mCherry cells (2.5 × 10^5^ tumor cells s.q. in 100 μL (1:1 DPBS&Matrigel) were injected subcutaneously into the right flank of 6–8-week-old NSG-SGM3 mice. Tumor growth was monitored for 28 days using optical tomography (LumoTrace^®^ FLUO bioimaging system, Abisense, «Sirius» Federal Territory, Russia) on a weekly basis and caliper measurements twice a week. For bioluminescence imaging, mice were injected intraperitoneally with D-luciferin (150 mg/kg) 10 min prior to imaging. Imaging parameters were as follows: fluorescence—exposure 1200 ms, binning 1 × 1, LED 590, filter 650-40; bioluminescence—exposure 10,000 ms, binning 8 × 8, LED 0, no filter.

In addition, tumor dimensions were measured manually using digital calipers, and tumor volume was calculated using the formula: V = 4/3 × length × length × width.

Correlation between tumor volume measured by caliper and tumor area measured by optical tomography was analyzed using Pearson’s correlation coefficient.

### 4.4. Flow Cytometry Analysis

PBMCs were collected from naïve mice at the beginning of the experiment and tumor-bearing mice at the end of the experiment. PBMCs were isolated using density gradient centrifugation. Briefly, blood samples were collected in heparinized tubes (150 μL heparin + 450 μL PBS) and diluted with PBS to 3 mL. The diluted blood was carefully layered over 3 mL of Ficoll at a 1:1 ratio and centrifuged at 400× *g* for 20 min at 15 °C without acceleration or brakes. The PBMC layer was carefully collected and washed twice with 2% FBS in PBS (300× *g*, 7 min, 4 °C). Residual erythrocytes were lysed using ACK lysis buffer for 2 min, followed by a final wash. The isolated PBMCs were resuspended in PBS containing 2% FBS for further analysis. Cells were stained with Fixable Viability Dye-eFluor 780 (eBioscience, San Diego, CA, USA) and the following antibodies: anti-CD45-FITC (30-F11, BioLegend, London, UK), anti-CD11b-APC (M1/70, Invitrogen, Waltham, MA, USA), anti-Ly6C-PE-Cy7 (HK1.4, Invitrogen, Waltham, MA, USA), and anti-Ly6G-Pacific Blue (1A8, BioLegend, London, UK). Flow cytometry analysis was performed on a BD FACS Aria III and FlowJo software (BD Biosciences, San Jose, CA, USA). Monocyte-like population in NSG-SGM3 mice blood was identified as CD45^+^CD11b^+^Ly6C^+^Ly6G^-^, and neutrophil-like population was gated as CD45^+^CD11b^+^Ly6C^−^Ly6G^+^ (Appendix A).

### 4.5. Milliplex Cytokine Assay

Cytokine profiling of the MZT-Luc2-mCherry, glioblastoma (GBM), and acute myeloid leukemia (AML-193) cell lines was performed using a multiplex bead-based assay (MILLIPLEX^®^ Human Cytokine/Chemokine/Growth Factor Panel A 38 Plex Premixed Magnetic Bead Panel—Immunology Multiplex Assay) according to the manufacturer’s instructions. For this analysis, 1 × 10^5^; cells were seeded in 96-well plates in 100 μL of medium per well. Supernatants were collected 24 h after cell adhesion. Data were acquired on a Luminex MagPix system (Appendix A). Data processing was carried out using Belysa software v1.1.0 (Merck, Rahway, NJ, USA) at the Resource Center “Cell Technology and Immunology”, Sirius University of Science and Technology.

Serum samples and tumor lysates were analyzed for human and murine cytokines using multiplex bead-based assays (MILLIPLEX^®^ Human Cytokine/Chemokine/Growth Factor Panel A 38 Plex Premixed Magnetic Bead Panel—Immunology Multiplex Assay and MILLIPLEX^®^ MAP Mouse Cytokine/Chemokine Magnetic Bead Panel—Immunology Multiplex Assay) according to the manufacturer’s instructions. Data were acquired on a Luminex MagPix system (Appendix A). Z-score heatmaps were generated to represent cytokine profiles.

All cell lines used in this study were of human origin. Primary mesothelioma and glioblastoma cells were obtained from the Department of Thoracic Surgery and the Department of Neurosurgery at the Federal Research and Clinical Center, FMBA of Russia, respectively. The AML-193 cell line was purchased from the American Type Culture Collection (ATCC).

### 4.6. Real-Time PCR

Extraction of total RNA was carried out by the guanidinium thiocyanate–phenol–chloroform method. In summary, cells from tumor tissues were lysed with ExtractRNA reagent (Evrogen, Moscow, Russia). The tissues were snap-frozen in liquid nitrogen and homogenized using 1.4 mm ceramic beads (Qiagen, Valencia, CA, USA) in ExtractRNA buffer with PowerLyzer 24 homogenizer (Qiagen). The lysates were then mixed with 1-bromo-3-chloropropane (Sigma Aldrich, Burlington, VT, USA), incubated at room temperature for 3 min, and centrifuged at 12,000× *g* for 15 min at 4 °C. From these steps, the collected aqueous phase was subjected to RNA precipitation with isopropanol and 75% ethanol wash, air-dried, and dissolved in RNase/DNase-free water. The RNA concentration was assessed using the measurement taken from the Implen NanoPhotometer N50 Touch. The sodium salts of the isolated RNA were treated with DNAse I (Thermo Scientific, Waltham, MA, USA) to remove any contamination and cDNA synthesized with RevertAid reverse transcriptase (Thermo Scientific) using oligo(dT)18 according to the manufacturer’s specification. Samples of cDNA were diluted to 200 µL and frozen at −20 °C. For quantitative real-time PCR, qPCRmix-HS SYBR+LowROX (Evrogen) was used on Applied Biosystems 7500 Real-Time PCR System (Applied Biosystems, Waltham, MA, USA).

The following primers were used to assess gene expression in tumor lysates: *IL8*—Forward Sequence GAGAGTGATTGAGAGTGGACCAC, Reverse Sequence CACAACCCTCTGCACCCAGTTT; *ACTB*—Forward Sequence CACCATTGGCAATGAGCGGTTC, Reverse Sequence AGGTCTTTGCGGATGTCCACGT. For murine genes: *Ccl2*—Forward Sequence GCTACAAGAGGATCACCAGCAG, Reverse Sequence GTCTGGACCCATTCCTTCTTGG; *Il1b*—Forward Sequence TTTGACAGTGATGAGAATGACC, Reverse Sequence TGAGTGATACTGCCTGCC; *Cxcl10*—Forward Sequence CCAAGTGCTGCCGTCATTTTC, Reverse Sequence GGCTCGCAGGGATGATTTCAA; *Actb*—Forward Sequence CTCCTGAGGGCAAGTACTCTGTG, Reverse Sequence TAAAACGCAGCTCAGTAACAGTCC.

### 4.7. Histology and Immunohistochemistry

For hematoxylin and eosin (H&E) staining, tissue samples were fixed in a tenfold volume of 10% neutral buffered formalin for 24 h. The samples were then washed in PBS three times for 30 min each. Tissues were dehydrated through a graded series of ethanol (70%, 80%, and 96%), followed by isopropanol and xylene, before being embedded in paraffin. Sections were cut at 3–5 μm thickness using a rotary microtome and mounted on glass slides. Prior to staining, sections were deparaffinized in xylene, isopropanol, and ethanol. The slides were then stained with hematoxylin for 10 min, rinsed with distilled water followed by running tap water, and counterstained with eosin for 30 s. Finally, the sections were dehydrated through 96% ethanol, isopropanol, and xylene prior to being coverslipped using a mounting medium.

Immunoperoxidase staining of paraffin sections for Ki67 (Roche, clone 30-9, rabbit monoclonal), EMA (Roche, E29, mouse monoclonal), Calretinin (Roche, SP65, rabbit monoclonal), Pancytokeratin (Roche, AE1/AE3/PCK26 cocktail), WT1 (Roche, 6F-H2, mouse monoclonal), CEA (Roche, CEA31, mouse monoclonal), Ep-CAM(Roche, Ber-EP4, mouse monoclonal), and CD15 (Roche, MMA, mouse monoclonal) was performed in the Benchmark Ultra Immunostainer (Roche, Basel, Switzerland) using the manufacturer’s primary antibodies and the OptiView DAB IHC Detection Kit (Roche, Basel, Switzerland), in accordance with the manufacturer’s protocols. The stained and coverslipped slides were scanned using a Leica Aperio GT450 DX scanner (Leica Biosystems, Nussloch, Germany) and processed at 20× magnification with Aperio ImageScope software (Leica Biosystems, Nussloch, Germany).

For immunofluorescence staining, tumor samples were fixed in 10% formalin for 24 h, washed thrice in PBS for 30 min each, and cryoprotected in 15% then 30% sucrose solutions. Samples were then frozen, cryosectioned, mounted on slides, and fixed with acetone. After air-drying, sections were blocked with 200 μL of 50% FCS in mQ water for 2 h. The following antibodies and reagents were used for immunofluorescence staining: anti-Ly6G antibody (clone 1A8, BD Biosciences, San Jose, CA, USA) conjugated with FITC at 1:100 dilution to detect neutrophils; anti-CD11b antibody (clone M1/70, Thermo Fisher Scientific) conjugated with SB600 at 1:100 dilution to label macrophages; goat anti-rat secondary antibody (polyclonal, Abcam, Cambridge, UK) conjugated with AF488 at 1:250 dilution for subsequent validation; and MethylGreen nuclear stain (1:5000 dilution) to visualize cell nuclei. Antibodies were diluted in 10% FCS and applied to slides (250 μL/slide) for specific markers at appropriate concentrations. After 2 h incubation at 100 rpm, slides were washed with PBST and PBS, rinsed with mQ water, air-dried, and mounted. Prepared slides were stored at 4 °C. Images were obtained with Zeiss LSM 980, Zeiss Lab.A1 (Carl Zeiss Microscopy GmbH, Oberkochen, Germany) and using Zen 3.4 software (Carl Zeiss Microscopy GmbH, Oberkochen, Germany) for image capture and processing.

### 4.8. Statistical Analysis

Statistical analyses were performed using GraphPad Prism 10 software. Comparisons between two groups were made using paired or unpaired *t*-tests, as appropriate. Multiple group comparisons were analyzed by two-way ANOVA. Correlation analysis was performed using Pearson’s correlation coefficient. Cytokine profiling data were analyzed using the Mann–Whitney U test. *p* values < 0.05 were considered statistically significant. Gene expression was calculated using the 2^^-dCt^ method, where dCt represents the difference between the threshold cycle of the reference gene (*Actb* for murine genes and *ACTB* for human genes) and the target gene.

## 5. Conclusions

Our MZT-Luc2-mCherry xenograft model in NSG-SGM3 mice recapitulates key aspects of human mesothelioma progression, including characteristic cytokine production and myeloid cell recruitment, in the absence of adaptive immune responses. The distinct spatial organization of neutrophils and monocytes/macrophages within the tumor microenvironment provides new insights into the potential roles of these cells in supporting tumor growth and shaping the local immune landscape. This model provides a valuable platform for investigating the role of specific myeloid populations in mesothelioma development and for evaluating novel therapeutic strategies targeting the myeloid compartment; it also allows us to take another step towards more accurate understanding of PDX models of oncological diseases based on NSG-SGM3 mice, giving us the opportunity to see the basic interaction of the host organism (without a transplanted immune system) and the transplanted tumor. While this model may mimic certain aspects of the human tumor microenvironment more closely than conventional immunodeficient mouse strains, the absence of a complete immune system (its own or transplanted from humans) should be considered when interpreting the results and translating the findings to human disease.

## Figures and Tables

**Figure 1 cells-13-02135-f001:**
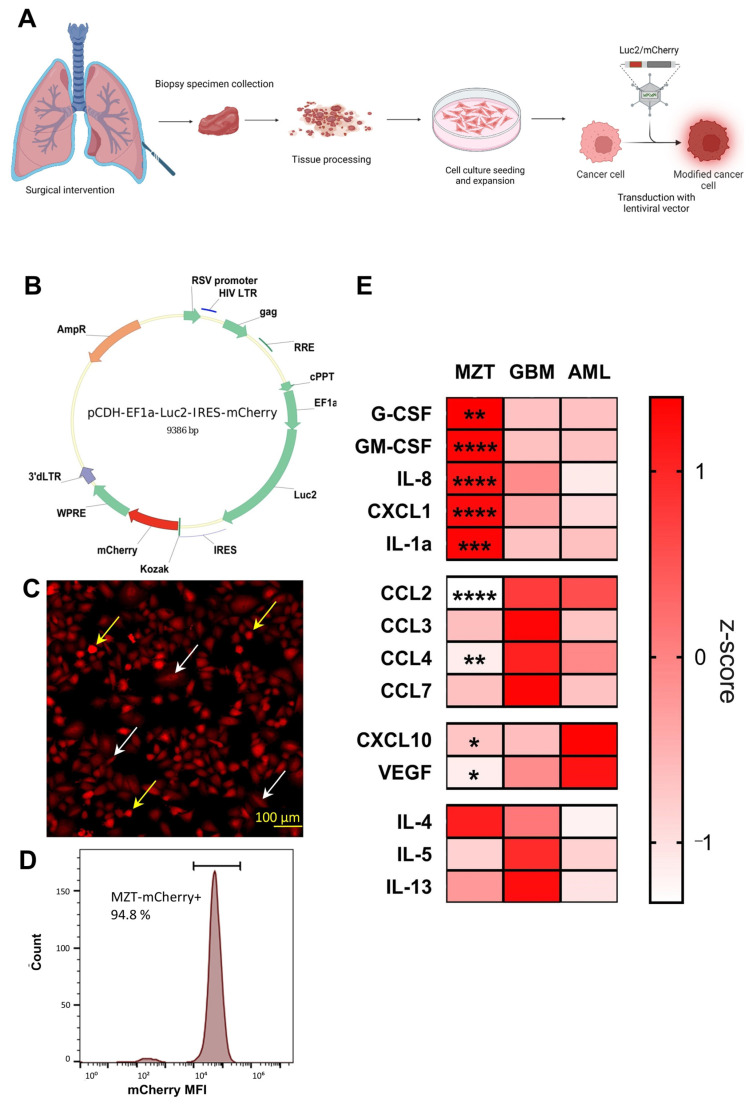
**Generation and cytokine profiling of primary mesothelioma reporter cell line.** (**A**) Scheme of primary mesothelioma isolation and lentiviral transduction. Primary tumor cells were obtained from a patient’s biopsy with malignant pleural mesothelioma. Tumor mass was dissociated mechanically to generate single-cell suspension. Cells were passaged 15 times and transduced with (**B**) pCDH-EF1a-Luc2-IRES-mCherry lentiviral vector, encoding reporter proteins—Luc2 and mCherry. (**C**) Cells in culture displayed elongated spindle-shaped morphology (white arrows) and anchorage-independent growth (anoikis resistance) (yellow arrows) (**D**) Mean-fluorescence intensity (MFI) of the established primary mesothelioma reporter cell line. (**E**) 1 × 10^5^ cells were seeded in 96-well plates in 100 μL of medium per well. Supernatants were collected 24 h after cell adhesion. Comparative human cytokine profiling of suspensions of three tumor cell lines: mesothelioma—MZT-Luc2-mCherry (MZT), glioblastoma (GBM), and acute myeloid leukemia (AML) using Z-score. Statistical analysis was performed using two-way analysis of variance (ANOVA). n = 3 samples per group. * for *p* < 0.05, ** for *p* < 0.01, *** for *p* < 0.001, **** for *p* < 0.0001; Z-score transformed concentration (pg/mL) in medium. Measurements were performed using a multiplex assay on cell culture supernatants.

**Figure 2 cells-13-02135-f002:**
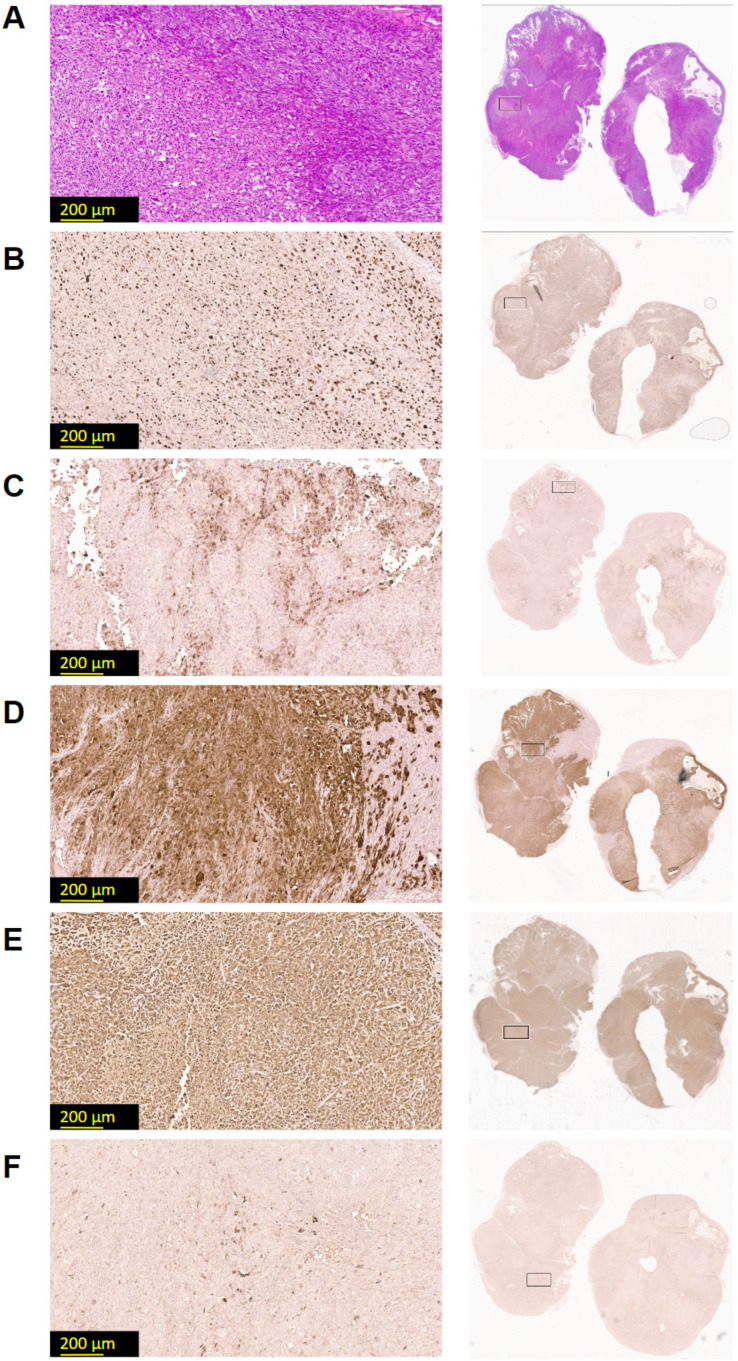
**Histology and immunohistochemistry of engrafted mesothelioma.** Used stainings: (**A**) hematoxylin–eosin; (**B**) Ki-67; (**C**) Epithelial Membrane Antigen; (**D**) Calretinin; (**E**) Pancytokeratin; (**F**) Mesothelin. Histopathological examination revealed a neoplastic lesion consistent with epithelioid mesothelioma, exhibiting a predominantly solid growth pattern with focal areas of necrosis (n = 4). The images were obtained by Leica Aperio GT450 DX scanner (Leica Biosystems, Deer Park, TX, USA) and then processed at 20× magnification using Aperio ImageScope 12.4.6 software.

**Figure 3 cells-13-02135-f003:**
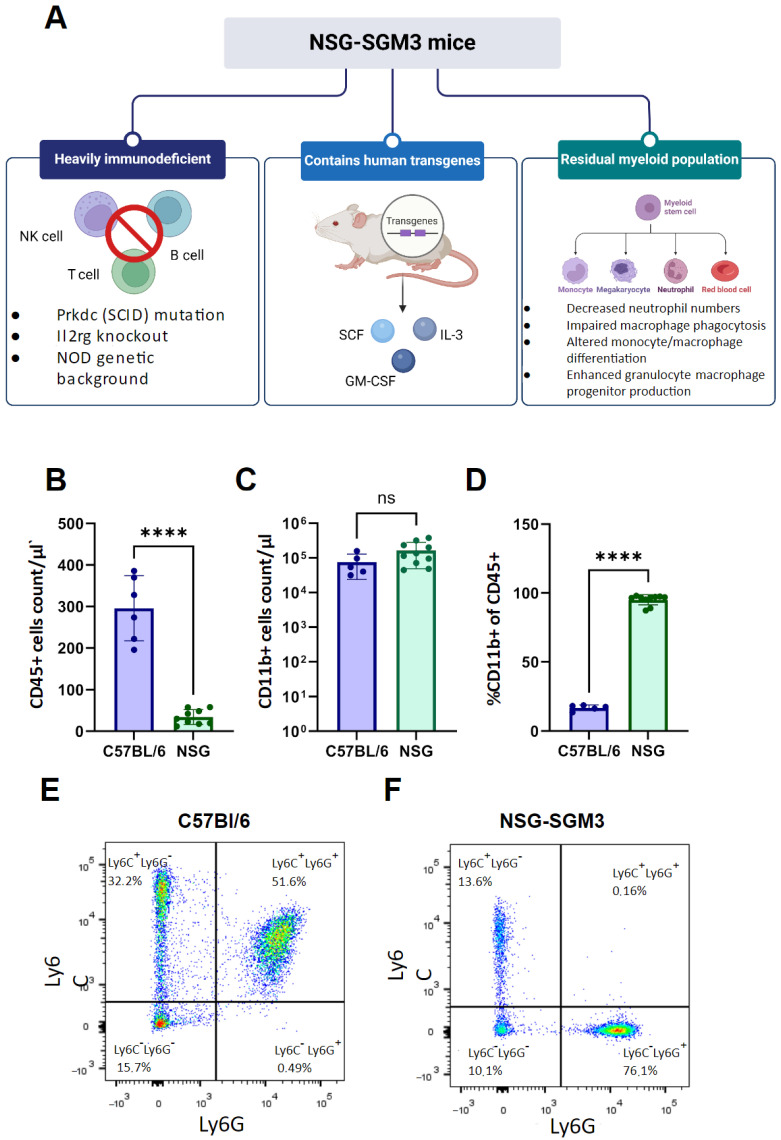
**Immunodeficient NSG-SGM3 mice possess residual myeloid populations.** (**A**) Summary of inherent NSG-SGM3 (NOD.Cg-Prkdcscid Il2rgtm1Wjl Tg (CMV-IL3,CSF2,KITLG)1Eav/MloySzJ) mouse strain traits. (**B**) Comparison of CD45^+^ cell counts between C57BL/6 and NSG-SGM3 mice. NSG-SGM3 mice show significantly lower CD45^+^ cell counts per μL (*p* < 0.0001). (**C**) Comparison of CD45 + CD11b^+^ cell counts per μL between C57BL/6 and NSG-SGM3 mice. No significant difference was observed. (**D**) Comparison of CD11b^+^ cells as a percentage of CD45^+^ cells between C57BL/6 and NSG-SGM3 mice. NSG-SGM3 mice exhibit a significantly higher percentage of CD11b^+^ cells (**** for *p* < 0.0001). Statistical analysis (**B**–**D**) was performed using paired *t*-test. (**E**) Representative flow cytometry dot plot of myeloid populations in C57BL/6 mice. (**F**) Representative flow cytometry dot plot of myeloid populations in NSG-SGM3 mice. Cells (in (**E**,**F**)) were gated from VD⁻CD45^+^CD11b^+^ population.

**Figure 4 cells-13-02135-f004:**
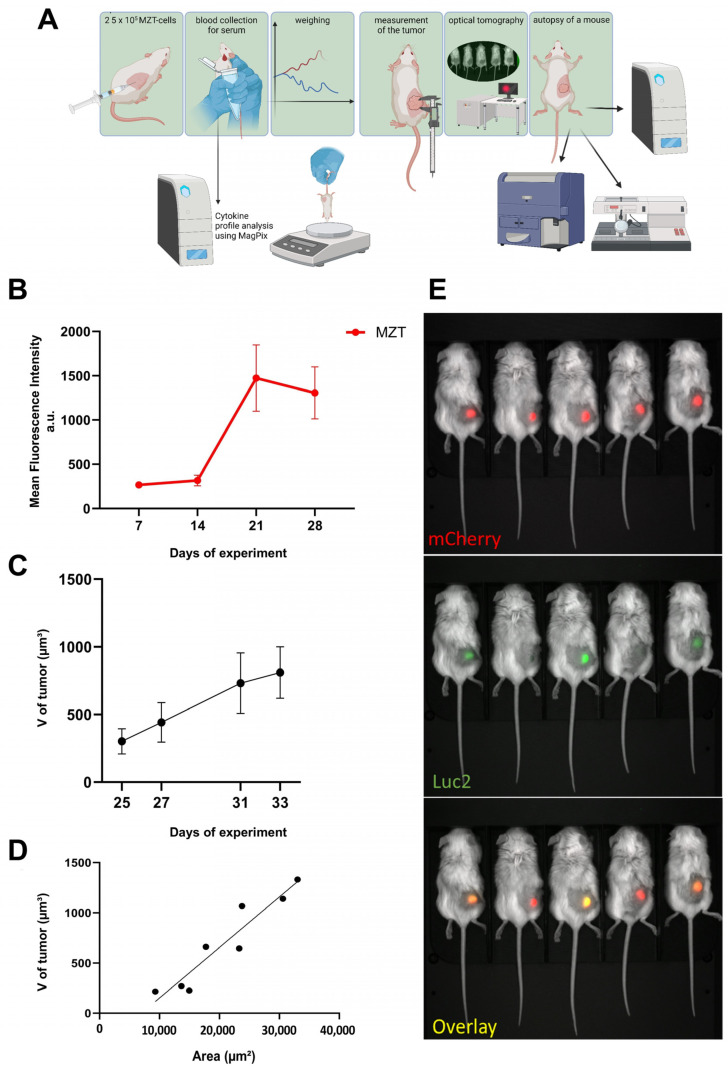
**MZT-Luc2-mCherry successful engraftment in NSG-SGM3 mice provides a useful model of tumorigenesis.** (**A**) Schematic representation of the experimental timeline and methods used. (**B**) Mean fluorescence intensity of the tumors as measured by optical tomography using LumoTrace^®^ FLUO bioimaging system (Abisense, «Sirius» Federal Territory, Russia) over 28 days of tumor growth (n = 17). Data are presented as mean ± SEM and measured in arbitrary units (a.u.). (**C**) Tumor volume progression as measured by caliper over the course of the experiment (n = 5). Data are presented as mean ± SEM. (**D**) Correlation between tumor volume measured by caliper and tumor area measured by optical tomography using LumoTrace^®^ FLUO bioimaging system. Correlation analysis performed using Pearson’s correlation coefficient (r). (**E**) Representative optical tomography images. Top: fluorescence measurement (exposure 1200 ms, binning 1 × 1, LED 590, filter 650-40). Middle: bioluminescence measurement (exposure 10,000 ms, binning 8 × 8, LED 0, no filter). Bottom: overlay image (bright field: exposure 1200 ms, binning 1 × 1, LED white light, no filter).

**Figure 5 cells-13-02135-f005:**
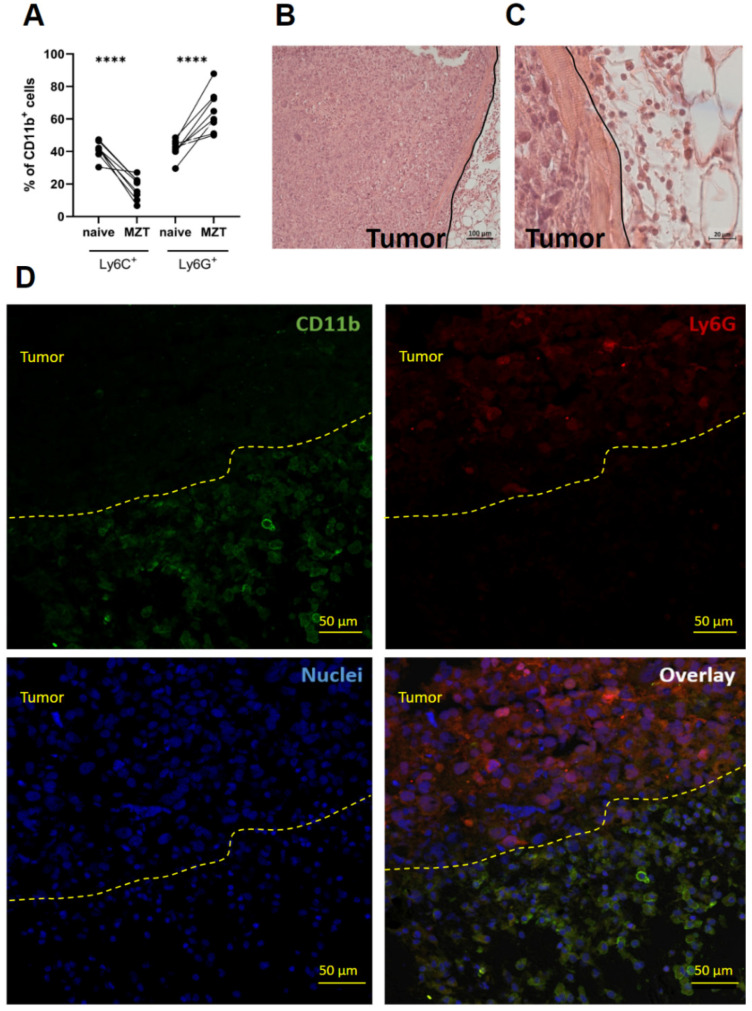
**Marked increase in the neutrophil-like population in the blood and at the tumor site of NSG-SGM3 mice with the engrafted primary mesothelioma.** (**A**) Paired comparison of Ly6C^+^ and Ly6G^+^ populations as a percentage of CD11b^+^ cells for each individual mouse at the beginning and end of the experiment. Data show a consistent decrease in Ly6C^+^ cells and an increase in Ly6G^+^ cells following mesothelioma engraftment. Comparison of Ly6C^+^ and Ly6G^+^ populations as a percentage of CD11b^+^ cells in naïve and mesothelioma-bearing mice. Statistical analysis was performed using paired *t*-test (**** for *p* < 0.0001). (**B**,**C**) Microscopic images of histological specimens stained with hematoxylin and eosin. These images showcase the tumor margins and the surrounding loose fibrous connective tissue. A black line has been added to approximate the border between the tumor and the adjacent tissue. Scale bars: 100 μm for (**A**), 20 μm for (**B**). (**D**) Confocal immunofluorescence microscopy analysis of CD11b^+^ and Ly6G^+^ cells in tumor tissue and CD11b^+^ cell distribution in and around the tumor microenvironment. Immunofluorescence staining for CD11b (green) on the periphery of the tumor and Ly6G (red) inside the tumor. Although CD11b signal intensity appears relatively low inside the tumor, the Ly6G^+^ cells are CD11b-positive, which is consistent with the expected phenotype of polymorphonuclear neutrophils. The tumor is shown up from the dashed line. Individual CD11b^+^ cells were detected within the tumor mass. Scale bars: 50 μm. Nuclei are counterstained with Methyl Green (blue). Images are representative of n = 6 independent samples.

**Figure 6 cells-13-02135-f006:**
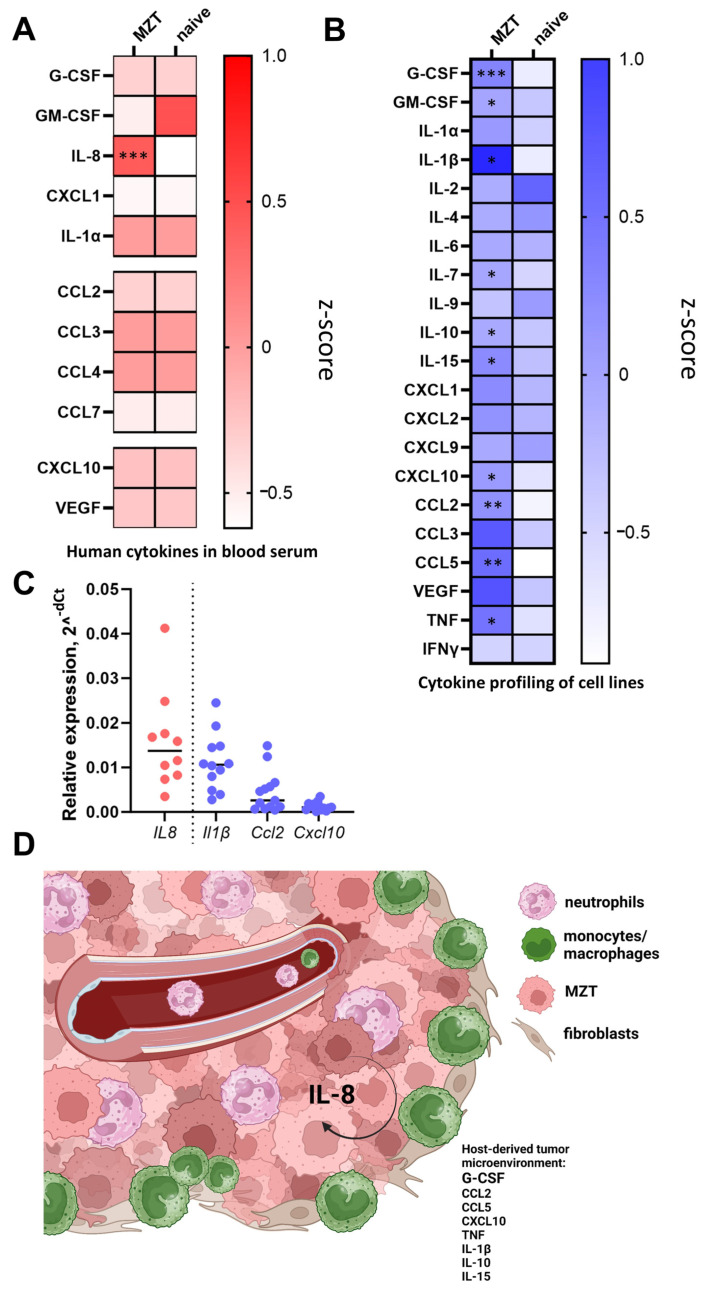
**Systemic cytokine production in tumor-bearing mice suggests a cross-talk between engrafted tumor cells and residual myeloid cells in NSG-SGM3 mice.** Z-score heatmaps representing cytokine profiles in NSG-SGM3 mice of 2 groups: naïve and 4 weeks after MZT-Luc2-mCherry cell line engraftment. (**A**) Human and (**B**) murine cytokine profiles in blood serum. Statistical analysis (**A**,**B**) was performed using the Mann–Whitney U test (* for *p* < 0.05, ** for *p* < 0.01, *** for *p* < 0.001). MZT—n = 10, naïve—n = 6; Z-score transformed concentrations (pg/mL) in serum. Measurements were performed using multiplex assay of blood serum samples (**A**,**B**). (**C**) The analysis for inflammatory human (*IL8*) and murine (*Il1b*, *Ccl2*, and *Cxcl10*) gene expression in the engrafted tumors by real-time PCR. Relative gene expression is shown as a 2^^-dCt^ normalized by human or mouse beta-actin gene, respectively (n = 13). (**D**) Schematic representation of the complex interplay between mesothelioma tumor cells, tumor microenvironment, and cytokine signaling in NSG-SGM3 mice.

## Data Availability

The data that support the findings of this study are available from the corresponding author upon reasonable request.

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
