# Peer review of "Myeloid Cell Mobilization and Recruitment by Human Mesothelioma in NSG-SGM3 Mice"

_cells, 2024, doi:10.3390/cells13242135_

Round 1
Reviewer 1 Report
Comments and Suggestions for Authors
This manuscript presents an innovative human mesothelioma model using a primary cell line derived from human pleural biopsy implanted in NSG-SGM3 mice. Mesothelioma cells secrete significantly increased levels of G-CSF, GM-CSF, IL-8, CXCL1, and IL-1α, stimulating neutrophil mobilization and supporting tumor progression. The increase in Ly6G+ neutrophil-like cells and the reduction in Ly6C+ monocytes in tumor-bearing mice suggest systemic effects of mesothelioma on the host immune system. Tumor macrophages adopt an immunomodulatory (M2) phenotype, enhancing pro-tumor effects, and the infiltration of CD11b+ cells at the tumor periphery indicates the formation of an immunosuppressive environment that facilitates immune evasion. The model can be used to study the interactions between tumor cells and the myeloid compartment, which is relevant in developing immuno-oncology therapies.
The article can be published but with minor changes/additions.
What type of medium was used to grow and passage the cells? This may influence the morphology and anchorage‐independent growth type. What was the rate/number of cells per passage?
Page 4, lines 152-153: “Measurements were performed using a multiplex assay on cell culture supernatants collected 24 hours after culturing 1 x 10^5 cells in 100 μL of medium.” Is this correct? On what surface were 10^5 cells cultured.
Author Response
Q1: What type of medium was used to grow and passage the cells? This may influence the morphology and anchorage‐independent growth type. What was the rate/number of cells per passage?
R1: Cells were maintained in DMEM/F12 medium supplemented with 10% FBS and antibiotic-antimycotic solution using 75 cm² treated flasks. Primary tumor cells were passaged 20 times, they were further passaged 5 times after transduction. Our standard passaging protocol involved harvesting 2×10⁶ cells and reseeding 2×10⁵ cells for subsequent passages. When retrieving cells from cryopreservation, we seeded 1×10⁶ cells and monitored them until monolayer formation before establishing the regular passaging schedule. The passaging frequency was weekly during the first week of culture, then increased to every two days as cells reached optimal growth conditions. Experiments were performed 2-3 passages after cryopreservation. We added this information in lines 416-422, 425-426.
Q2: Page 4, lines 152-153: “Measurements were performed using a multiplex assay on cell culture supernatants collected 24 hours after culturing 1 x 10^5 cells in 100 μL of medium.” Is this correct? On what surface were 10^5 cells cultured.
R2: Thank you for pointing this out. We would like to clarify that the experiments were conducted in 96-well plates. Specifically, 1 × 10⁵ cells were seeded per well in 100 μL of medium. The supernatants were collected 24 hours after initial cell adhesion for subsequent multiplex analysis. We added this information in lines 479-480.
Reviewer 2 Report
Comments and Suggestions for Authors
The study Shindyapin et al. provides a new PDX model of mesothelioma, well-described experiments, and data analysis of myeloid-derived cell content in blood samples and tumor microenvironment. The data is solid, the manuscript is well-designed, and is of great interest to readers. The manuscript revealed an innovative interplay between PMN cells in blood and tumor microenvironment. Tumor-bearing mice showed a marked increase in the neutrophil-like Ly6G+ population in both blood and tumor microenvironment; while a decrease in the monocyte-like Ly6C+ population within the CD11b+ compartment in the blood. The detailed characterization of cytokine/chemokine levels in blood versus tumor microenvironment and their impact on myeloid-derived cell composition and tumor progression is described. I recommend the manuscript for publication in Cells MDPI journal with minor revision. Please see below a couple of minor recommendations:
1. It is expected that PMN cells are CD11b+Ly6G+ as shown by authors using flow cytometry data. Fig. 5D exhibits a very low CD11b signal for Ly6G+ cells in the tumor core. It will be helpful if authors clarify for readers in the text that these Ly6G+ cells are CD11b positive (although with low intensity).
2. Fig. 1D. Please add % after 94,8 in the plot.
3. Line 448. Please explain how PBMCs (blood samples) were collected.
4. Supplementary Table 1 legends. Please include sources of GBM and AML cells.
Author Response
Q1: It is expected that PMN cells are CD11b+Ly6G+ as shown by authors using flow cytometry data. Fig. 5D exhibits a very low CD11b signal for Ly6G+ cells in the tumor core. It will be helpful if authors clarify for readers in the text that these Ly6G+ cells are CD11b positive (although with low intensity).
Response 1: Thank you for this very valuable observation. Indeed, we have added clarification in the text regarding the CD11b expression status of Ly6G+ cells in the tumor core (lines 287-288).
Q2: Fig. 1D. Please add % after 94,8 in the plot.
R2: Done
Q3: Line 448. Please explain how PBMCs (blood samples) were collected.
R3: We added a detailed description of our PBMC isolation protocol to the Methods section (lines 458-465). We used a standard density gradient centrifugation method with Ficoll, including collection of the mononuclear cell layer and subsequent washing steps. Erythrocyte contamination was eliminated using ACK lysis buffer, and cells were maintained in PBS supplemented with 2% FBS.
Q4: Supplementary Table 1 legends. Please include sources of GBM and AML cells.
R4: We added detailed information to the Materials and Methods section in the revised version (lines 475, 491-495). Mesothelioma and glioblastoma cell lines were established for the primary tumor mass of patients at the Department of Thoracic Surgery and the Department of Neurosurgery, Federal Research and Clinical Center, FMBA of Russia, respectively. AML-193 cell line from the American Type Culture Collection (ATCC).